# Biomedical Variables and Adaptation to Disease and Health-Related Quality of Life in Polish Patients with MS

**DOI:** 10.3390/ijerph15122678

**Published:** 2018-11-28

**Authors:** Joanna Dymecka, Mariola Bidzan

**Affiliations:** 1Institute of Psychology, University of Opole, 45-052 Opole, Poland; jdymecka@uni.opole.pl; 2Institute of Psychology, University of Gdansk, 80-309 Gdansk, Poland

**Keywords:** chronic illness, disability, acceptance of illness, neurology, mood disorders, fatigue, quality of life

## Abstract

The aim of this research was to assess the level of adaptation to multiple sclerosis (*Sclerosis multiplex*; MS) and health-related quality of life (HRQoL) of the study population as well as to determine the relationship between biomedical factors related to the course of multiple sclerosis, adaptation to the disease, and HRQoL. Analysis of medical records, clinical and psychological interviews, the Extended Disability Status Scale (EDSS), Guy’s Neurological Disability Scale (GNDS), the Acceptance of Illness Scale (AIS), and the Multiple Sclerosis Impact Scale 29 (MSIS-29) were collected from 137 patients with MS. It was found that there was a relation between motor impairment, neurological disability, adaptation to illness, and HRQoL; it was also found that there were negative correlations between adaptation to illness and the severity of lower-limb disability, fatigue, mood disorders, other problems related to MS, and upper-limb disability. Of all the symptoms, lower-limb disability, fatigue, and mood disorders had the strongest relation with adaptation. All of the analysed symptoms were found to correlate with HRQoL. Of all the symptoms, HRQoL was most affected by lower- and upper-limb disability, fatigue, other MS problems, and mood disorders.

## 1. Introduction

Multiple sclerosis (*Sclerosis multiplex*, MS) is a chronic, progressive, autoimmune central nervous system disease which affects the physical, mental, and social functioning of an individual [1,2]. It is one of the most common neurological disorders in young adults and the most common nontraumatic cause of disability among young and middle-aged individuals [3,4,5]. By destroying myelin sheaths and through axon degeneration in the brain and spinal cord, MS leads to permanent disability and its course is unpredictable and highly varied.

The clinical manifestation of the disease is related to many neurological disorders, such as mobility problems, sensory and vision disturbances, sphincter disorders, fatigue, cognitive disorders, and mood disorders, which lead to the gradual development of disability. MS has a relatively small effect on life expectancy, but its long duration and progression mean it has a significant impact on the functioning of an individual. MS can appear in people who have never had health problems before and the uncertain prognosis and risk of disability can significantly affect their mental health as they enter adulthood and start planning their careers and family life [6]. One of the toughest aspects of MS is its unpredictable course, which can be even more devastating than the symptoms themselves. MS may have a relapsing–remitting or chronically progressive form and may also be mild or acute, as in the case of the Marburg variant.

The heterogeneity of MS, the unpredictability of exacerbations, the lack of effective drugs, and the prospect of disability make people perceive their illness as threatening and changeable [7,8]. Uncertainty, so characteristic of MS, is one of the major challenges in adapting to the disease. It is a major source of stress in patients and is associated with increased risk of emotional problems, depression, and psychological stress, and it also affects an individual’s adaptation to the disease [1,9].

Psychological adaptation to chronic health loss can be understood as optimal functioning of an individual given their circumstances. This includes emotional, cognitive, and behavioural functioning. The adaptation process is a gradual transition from the perception of a medical condition or disability as a catastrophe to its acceptance and treating it as just another personal feature of an individual. The goal of the adaptation process is to minimise the influence of the medical condition on the patient’s life, helping them cope with negative emotions and accept the changes they experience. At the same time, the levels of acceptance of an illness may be considered an indicator of adaptation to the illness and the limitations that come with it [10,11,12].

Therefore, MS is considered to be a disease that strongly influences health-related quality of life (HRQoL), which poses a serious challenge to mental adaptation [13,14]. Previous studies have shown that HRQoL is significantly worse in patients than in sex- and age-matched general population controls because MS affects all dimensions of human functioning [15,16,17,18,19,20,21,22,23,24,25,26,27,28,29,30,31,32,33]. The greatest differences in the quality of life between patients with MS and healthy individuals are related to physical health and physical limitations, while there are smaller differences in terms of pain and emotional well-being [34]. In addition, the satisfaction with life of people suffering from MS is significantly lower than of those with other chronic diseases, such as inflammatory bowel disease, rheumatoid arthritis, epilepsy, diabetes, or cardiovascular disease [16,35,36,37,38,39,40,41]. The significant decrease in HRQoL in this population is primarily influenced by: disease onset in the most productive years of an individual’s life, which affects personal development and future plans negatively and jeopardizes their autonomy, independence, and dignity; lack of effective treatment; the unpredictable course of the disease (it is difficult to determine when recurrences will happen and how serious the resulting disability will be); and the wide range of symptoms [16,37,42,43]. All this makes it difficult for patients to maintain control over their symptoms and their life. There is the additional burden of neuropsychiatric complications, including cognitive disorders and mood disorders such as depression, which may be manifestations of the disease itself (i.e., demyelinization and inflammation) or may be related to the process of struggling with the symptoms of the disease and its unpredictability. In the Polish population, an important problem is also the limited availability and high cost of treatments which modify the course of the disease. Poland is the only country in the European Union that has not introduced a national MS treatment program, so patients have little access to free treatment and rehabilitation [44]. However, studies show that the quality of life of Polish MS patients does not differ significantly from the quality of life of those in other countries, such as the United States or Italy [45].

## 2. Materials and Methods

### 2.1. Research Aims

The purpose of this research was:to assess the level of adaptation to disease and health-related quality of life of the study population; andto determine the relationship between biomedical factors related to the course of multiple sclerosis, adaptation to the disease, and health-related quality of life.

### 2.2. Research Procedure

The research was carried out in the years 2013–2016. Consent was granted by the Ethics Committee at the Institute of Psychology of the University of Gdansk, Poland (No. 19/06/2015). The group studied consisted of patients diagnosed with MS who were on rehabilitation stays at the John Paul II Rehabilitation Centre for Individuals with Multiple Sclerosis in Borne Sulinowo, as well as people under the care of the association of MS Patients in Głogów and the Twardziele group (located in the Gdansk–Gdynia–Sopot Tricity area). Patients with cognitive deficits which impeded the understanding of psychological questionnaires were excluded from the study (i.e., patients who scored more than 3 points on the Cognitive Disorders subscale of Guy’s Neurological Disability Scale (GNDS) questionnaire). The study was usually conducted in a single meeting with the patient, with no time limit; the duration was adjusted to the psychophysical capacity of the respondents. Patients were asked to consent to participate in the study before it began. All patients agreed to participate in the study, which was preceded by a short conversation on general topics aimed at reducing anxiety. The study consisted of the completion of a set of questionnaires which were always presented to the respondents in the same order.

### 2.3. Characteristics of the Study Population

The group consisted of 137 individuals diagnosed with multiple sclerosis (53.3% women, 46.7% men). The slight preponderance of women may be due to the epidemiological characteristics of the disease and its prevalence in females. The mean age of the examined patients was 46.47 (SD = 12.59), the youngest was 18 years old and the oldest was 73 years old. Most were aged over 50 years (44.5%), 24.1% were between 31 and 40 years of age, 19% were between 41 and 50 years of age, and 11.7% were between 20 and 30 years of age. Only one person (0.7%) at the time of the examination was under 20-years old. The majority of respondents came from medium (50,000–150,000 inhabitants; 33.6%) and large (above 150,000 inhabitants; 28.5%) cities.

The group varied in terms of level of education. The majority of respondents had secondary education (42.3%), 27.7% had a master’s degree, 18.3% had vocational education, 9.5% had a bachelor’s degree, only two (1.5%) had basic education, and one (0.7%) was still in secondary school.

Most of the respondents were on a pension or retired (63.5%), 19.7% were professionally active, 15.3% did not work at all, and two (1.5%) were the sole bread winners.

The study population was diverse in terms of financial situation. Most respondents rated their financial situation as average (59.9%), 24.8% as good, 12.4% as difficult, 2.2% as very difficult, and 0.7% as very good.

### 2.4. Research Methods

The following research methods were used:Analysis of medical records.Clinical and psychological interviews.Questionnaires for individuals with multiple sclerosis.The Extended Disability Status Scale (EDSS) by Kurtzke is the most commonly used and most popular scale for assessing levels of disability in individuals affected by multiple sclerosis. The scale includes 20 levels of disability; however, in order to make it consistent with the scoring on the older version of the scale (Disability Status Scale, DSS), half-points were introduced. Higher scores on the scale indicate higher levels of disability [46,47].Guy’s Neurological Disability Scale (GNDS) assesses disability and symptoms experienced by individuals with multiple sclerosis. It consists of 12 subscales regarding problems in various areas of functioning: cognitive, mood, vision, speech, swallowing, upper-limb function, lower-limb function, bladder function, bowel function, sexual function, fatigue, and others. In each subscale, disability is assessed on six levels of severity. Results on separate subscales are summed in order to describe the overall levels of disability of a patient. The higher the score, the greater the disability [48].The Acceptance of Illness Scale (AIS), constructed by Felton, Revenson, and Hinrichsen in 1984, adapted to Polish by Juczyński [49], assesses a patient’s adjustment to the limitations caused by the condition. It consists of eightstatements describing consequences of poor health. Respondents assess each statement on a 5-level scale (1—I fully agree, 5—I fully disagree). Low scores indicate a lack of acceptance of the condition and a strong sense of psychological discomfort. High scores indicate acceptance of the condition and a lack of negative emotions associated with it. The higher the acceptance of the condition, the better the adjustment for it. The reliability of the Polish version of the scale is satisfactory, the Cronbach α coefficient is equal to 0.85.The Multiple Sclerosis Impact Scale 29 (MSIS-29) by Hobart and Thompson was adapted to Polish by Jamroz-Wiśniewska, Papuć, Bartosik-Psujek, Belniak, Mitosek-Szewczyk, and Stelmasika [50]. The scale consists of 29 questions: 20 regarding one’s physical condition and 9 regarding one’s psychological condition. Participants assess each of the items on a 5-level Likert scale. The higher the score, the higher the impact of multiple sclerosis on one’s quality of life. An overall score as well as scores on particular subscales can be calculated. Reliability and validity of the Polish version of the scale are satisfactory. Cronbach α coefficients were equal to 0.97 for the physical factor of quality of life and 0.94 for the psychological factor of the quality of life.

## 3. Results

### 3.1. Characteristics of Biomedical Factors Associated with the Course of Multiple Sclerosis in the Studied Population

Multiple sclerosis is a clinically heterogeneous condition, and in its course, it can take several forms, the three main ones being: relapsing-remitting, primary-progressing, and secondary-progressive multiple sclerosis. In the studied population, the largest group was patients with relapsing–remitting multiple sclerosis (RRMS; 31.4%). This is associated with the fact that this form is exhibited the most frequently in the natural course of the disease. The second largest group were patients with primary-progressive multiple sclerosis (22.6%), and third was secondary-progressive multiple sclerosis (16.1%). The smallest group was affected by the progressive-relapsing form (5.8%). A significant fraction of the participants (24.1%) were not assessed with regards to the form of their multiple sclerosis. The mean duration of the condition in the studied population was 14.61 years (SD = 8.31). The shortest duration of the condition in the studied population was several months and the longest was 42 years. The mean age at which the participants were diagnosed with the condition was 33.94 years (SD = 10.65), the lowest age at the time of the diagnosis was 15 years and the highest 61 years.

Practically all types of symptoms associated with central nervous system damage can occur in the course of multiple sclerosis; however, most studies classify patients based on the level of their motor disability. Motor disability in the studied population was assessed using the EDSS scale. The mean level of disability in the studied population was equal to 4.57 (SD = 2.10). The lowest score for disability was equal to 0 (i.e., neurological state was normal) and the highest score was equal to 9, indicating complete motor disability characteristic of bed-bound patients who can communicate with those around them. Individuals with mild motor disability were the biggest group (EDSS 0–4, 43.8%), followed by individuals who required unilateral or bilateral assistance when walking (EDSS 6–6.5,19.0%) and individuals with moderate disability (EDSS 4.5–5.5,18.2%). Individuals with significant disability were the smallest group, including individuals restricted to a wheelchair (EDSS 7–7.5,14.6%) and individuals with significant limitations in their ability to care for themselves, including bed-bound patients (EDSS 8–9.5,4.4%).

As well as the assessment of levels of disability using the EDSS scale, participants were asked to assess their ability to move on their own. A total of 68.6% of participants declared the ability to move on their own and 31.4% declared being unable to walk on their own. The need to use a mobility aid was also assessed. A total of 58.5% of participants declared the need to use a mobility aid and 41.6% declared no need for using such equipment. Among the individuals who declared the need to use a mobility aid, 25.6% used crutches, 20.4% used a wheelchair, 4.4% used a walking frame, and 8% used some other equipment.

Apart from mobility problems, disability associated with multiple sclerosis may also concern other areas of functioning. The GNDS scale was used to assess this type of disability, hereafter referred to as neurological disability. This scale, apart from measuring lower-limb functions, also measures cognitive functioning, mood, vision and speech functions, ability to swallow, upper-limb function, bladder function, bowel function, sexual function, fatigue, and “other problems”. The mean result on this scale, on which it is possible to score between 0 and 60, was 16.98, with the lowest and highest scores equal to 0 and 37, respectively. The most severe symptom in this group was fatigue (2.86), followed by problems with bladder function (2.64), lower-limb function (2.16), and sexual functioning (2.08). Among the less frequent symptoms were other problems (1.79), mood problems (1.47), upper-limb disability (1.36), bowel function problems (1.07), and cognitive function problems (1.04). The least frequent problems concerned swallowing (0.36), speech (0.49), and vision (0.52).

Though there is currently no available method for curing multiple sclerosis, there are some therapies which can modify the course of the condition and slow down its progression. In the studied group, 62.04% had access to such therapies and 37.96% never used such types of treatment. Interferon beta was the most commonly used medication in the studied population (43.07%), followed by mitoxantrone (16.79%) and glatiramer acetate (7.30%). All patients with the RRMS form of the condition were not undergoing a relapse at the time and were not using steroids.

### 3.2. Adaptation to the Disease

Adaptation to disease was investigated with the AIS. The mean acceptance of illness in the study population was 24.20 (SD = 8.55), indicating the respondents were adapted to MS to an average extent. The lowest score obtained by a patient on the AIS was 8, and the highest was 40.

### 3.3. Health-Related Quality of Life in Individuals with Multiple Sclerosis

HRQoL in the study population was determined using the MSIS-29. The results are presented in Table 1.

As well as the general level of quality of life, physical and mental aspects were determined. The mean for the physical dimension was 51.62. This result was higher than that in the validation analysis of the Polish version of the scale, which indicates that the assessment of the physical aspect of quality of life by the examined subjects was lower. The lowest score in the examined population was 20, the highest was 97. The mean for the mental aspect of the quality of life of the subjects was 23.74, which was lower than that in the validation study. This indicates that the respondents assessed the mental aspect of their quality of life better.

### 3.4. Relation of Biomedical Variables to Adaptation to Illness and Health-Related Quality of Life

It was assumed for the purposes of the study that biomedical variables can influence adaptation to illness and HRQoL. Therefore, it was examined whether the illness is correlated with adaptation to MS and HRQoL. An analysis of variance of differences was done between patients suffering from three types of multiple sclerosis (relapsing-remitting, primary-progressive, and secondary-progressive) concerning the investigated dependent variables—adaptation to illness and health-related quality of life. The preliminary analysis using Levene’s test showed no significant deviations from the assumption of the equality of variance.

Tukey’s posthoc test showed significant differences only in the impact of MS on HRQoL and its physical aspect. Patients with the primary-and secondary-progressive illness had higher scores than patients with the relapsing-remitting type (see Table 2). This means that among patients with the progressive types of the illness, its impact on HRQoL, including its physical aspect, wasgreater than in the case of the relapsing-remitting type.

The relation between biomedical variables, such as the duration of illness, age at the time of diagnosis, degree of disability as measured with the EDSS and GNDS, and adaptation to illness and HRQoL, was determined with Pearson’s linear correlation method and is presented in Table 3.

There was no relation between the duration of illness, age at the time of diagnosis, adaptation to illness, and HRQoL. However, it was found that there was a relation between motor impairment (EDSS), neurological disability (GNDS), adaptation to illness, and HRQoL. There was a statistically significant moderate negative correlation between adaptation to illness and both degree of motor impairment and neurological disability, which means that the greater the disability, the worse the adaptation to illness. There was also a statistically significant positive correlation between HRQoL and both degree of motor impairment and neurological disability. HRQoL was more closely related to neurological disability than to motor impairment. The relation between assessment of self-mobility, the need to use rehabilitation equipment to move, and the need for treatment which modifies the course of the illness versus adaptation to multiple sclerosis and HRQoL was also analysed. This relationship was evaluated using the point-biserial correlation analysis method and results are shown in Table 4.

A link between the need to use rehabilitation equipment to move and adaptation to illness was found. Those who did not need to use the equipment were better adapted to MS. There was no link between the ability to move independently, application of treatment, and adaptation to illness. An analysis of the relationships between the above variables and HRQoL revealed a relation between the assessment of self-mobility and the need for rehabilitation equipment and HRQoL. People who thought they were able to move on their own assessed their HRQoL and its physical aspect better. It was also found that people who moved independently assessed their HRQoL higher than those using crutches or wheelchairs. Additionally, no relation was observed between the application of treatment modifying the course of the illness and HRQoL and its two dimensions.

The relationship between MS symptoms, adaptation to illness, and HRQoL was analysed. This was evaluated using Spearman’s rank correlation method and is presented in Table 5.

It was demonstrated that there was a relation between the five symptoms of the illness and adaptation to it.

It was found that there were negative correlations between adaptation to illness and the severity of lower- and upper-limb disability, fatigue, mood disorders, and other problems related to MS. The greater the severity of the above symptoms, the worse the adaptation to multiple sclerosis.

All of the analysed symptoms were found to correlate with HRQoL. The greater the severity of the symptoms, the greater their influence on HRQoL. Of all the symptoms, HRQoL was most affected by lower- and upper-limb disability, fatigue, other MS problems, and mood disorders. The majority of multiple sclerosis symptoms (except impaired swallowing) were reported to correlate with the physical dimension of HRQoL. The greater the severity of the symptoms, the greater their influence on HRQoL. The physical dimension of HRQoL was most affected by lower- and upper-limb disability.

All eight symptoms examined were found to correlate with the mental dimension of HRQoL. The mental dimension of HRQoL was most affected by mood disorders, cognitive disorders, and fatigue.

## 4. Discussion

The relation between type of illness, adaptation, and health-related quality of life was first analysed. There was no difference in the degree of adaptation among patients with different types of MS. This is not consistent with the results of other studies, which found that type of illness affects the level of adaptation to multiple sclerosis [51]. Most likely, other variables have a more significant impact on the level of adaptation and disrupt this relation. Perhaps the level of adaptation is more influenced not by the way the illness progresses but by the rate at which the disability develops.

It has been shown, however, that patients with the progressive type of illness assess their quality of life worse. This is consistent with other studies [20,52,53,54]. However, there are studies that did not report a relation between the course of illness and quality of life [55]. Some researchers point out that HRQoL and adaptation to illness are more affected not by its course—whether it is relapsing or progressive—but by whether it is a mild or aggressive process, with the latter being associated with lower levels of HRQoL [16]. In other studies, it was found that the speed at which the illness progresses has a greater effect on quality of life during earlier stages of the illness [34]. It was shown that the high frequency of recurrence and high clinical activity of the illness at its onset affected HRQoL negatively in the long run [54]. In addition, HRQoL was found to be lower during relapse than during remission [56]. The present study did not find a relation between the duration of illness or age at the time of diagnosis with adaptation to multiple sclerosis and HRQoL. A significant number of studies indicate that long duration of illness, which is connected to its progression and the occurrence of serious symptoms, decreases HRQoL [29,57]. The results are not conclusive, as there are reports that duration of illness has only a moderate effect on HRQoL, and longer duration of illness is associated with decreased HRQoL [58]. On the other hand, there are studies which have shown that although the duration of the disease affects the degree of disability, it does not always entail a decrease of HRQoL [16]. This lack of correlation may be explained by the link between HRQoL and adaptation to MS, and adaptation may improve over time as the illness is most likely to occur in young individuals who are not yet mentally prepared to cope with serious disease and disability [15]. This thesis is confirmed by studies, according to which longer duration of illness and greater age are associated with better adaptation and higher HRQoL. Younger individuals with a shorter duration of illness who have mobility problems but still do not use wheelchairs most often assess their HRQoL negatively [55].

Next, the relations between adaptation to illness, HRQoL, motor impairment as measured by EDSS, and neurological disability assessed with GNDS were analysed. Motor impairment and neurological disability were found to be associated with adaptation to multiple sclerosis. The greater the disability, the worse the adaptation to illness. Other researchers also point to this link [51,59]. In addition, higher levels of disability are associated with higher levels of depression and frequent use of coping strategies focused on emotions and avoidance, which is a sign of poor adaptation to illness [60]. Limited capacity to deal with everyday problems and dependence on others make the illness harder to live with and, hence, harder to accept. This suggests a relation between adaptation to illness and the need to use rehabilitation equipment. This study also demonstrated the association of motor impairment and neurological disability with the physical and mental aspects of HRQoL; however, HRQoL was more closely related to neurological disability than to motor impairment. For the physical aspect of HRQoL, the relation was similar for both variables expressing disability, but the mental aspect correlated more strongly with neurological disability than with motor impairment. This is due to the fact that neurological disability involves a greater number of symptoms, including those of the psychological nature.

The strong negative correlation between the degree of physical and neurological disability and HRQoL has been shown in many studies [15,34,61,62,63,64,65,66,67,68,69,70,71,72,73]. This study also showed that people who think they are able to move on their own also assess their HRQoL and its physical aspect better. It was also found that people who move independently assess their HRQoL higher than those using crutches or wheelchairs, as confirmed in other studies [45]. Researchers also point out that HRQoL is lower in patients experiencing relapses and in those with progressing disability [74].

Some studies show that physical disability has a more negative impact on quality of life than mental problems [75]. However, others indicate that the relation between physical disability and HRQoL is not simple [55,76]. There is no linear and simple dependency between EDSS and HRQoL. Perhaps one of the factors influencing this relation is the ability to cope, which increases gradually with the duration of the illness. Some researchers believe that EDSS is a better predictor of quality of life at the onset of the illness than in its later stages, as patients gradually adapt to the illness [34]. The results indicate that the quality of life of MS patients is affected not only by EDSS but also the interaction of physical, psychological, and social factors [75], and disability is just one of many predictors [77]. It has also been suggested that the impact of disability on quality of life is stronger in men [32]. In addition, the increase in disability may adversely affect mental functioning, which in turn influences quality of life [78]. There are also studies which demonstrate that most aspects of quality of life are weakly related to disability [77], which means that physical disability is not always the main predictor of HRQoL. Patients, more often than professionals, point to the important role of other factors, such as fatigue, pain, or emotional problems, which affect their HRQoL [69]. More and more researchers indicate that disability can have only a minor impact on the HRQoL score of individuals with multiple sclerosis [16,79,80].

The results of this study contradict those that indicate that physical disability is only slightly related to adaptation to illness and HRQoL. The relationship between disability and quality of life in the current study is very strong, which is also confirmed by the analysis of the relation between the symptoms of multiple sclerosis, adaptation to illness, and quality of life. It has been shown that lower-limb disability is the symptom which has the greatest link to adaptation to illness, quality of life, and the physical aspect of HRQoL. The only lack of correlation was observed between lower-limb disability and the mental aspect of HRQoL.

Motor impairment is considered to be one of the predominant features of disability in MS, its primary marker, and one of the most obvious manifestations of the illness. Hence, it is strongly related to adaptation to illness and HRQoL. Lower-limb disability is associated with, inter alia, spasticity (which is related to gait abnormality, cramps, pain, and fatigue), one of the main causes of persistent disability in individuals with neurological diseases [81,82,83]. Spasticity negatively affects mobility and balance and is associated with significant discomfort, difficulty in daily functioning, and reduced HRQoL [84,85].

The clinical assessment of disability in MS is very often reduced to the functions of the lower limbs measured with EDSS. However, MS also includes invisible symptoms such as fatigue [80,86], which, as demonstrated in the current study, is strongly associated with adaptation to illness, HRQoL, and its two aspects. Fatigue is an important factor affecting physical and mental HRQol, even in patients at early stages of the disease [37,87]. It may be the first symptom of MS occurring before diagnosis [88]. Patients at every stage of the disease, even those with a low degree of disability, may experience significant fatigue [89], and 40%–50% of them say it is the worst symptom of the disease [90], more arduous than pain or even physical disability [17,20]. Fatigue is a major cause of incapacity for work and early retirement [80]. It can lead to the progression of the existing disability and negatively affect the outcome of rehabilitation [5]. It affects daily activity [80,91], negatively influencing quality of life regardless of physical disability [20] and depression [16]. A relation between fatigue and depressive symptoms in patients with MS was found, as fatigue exacerbates depression, and depression deepens fatigue [92]. These symptoms alter the assessment of HRQoL as strongly as physical symptoms such as motor disability [25,80].

As well as fatigue, other symptoms such as mood disorders, problems related to the course of multiple sclerosis, and upper-limb disability also correlate with adaptation to illness, HRQoL, and its two aspects. Mood disorders are related to adaptation to illness to a lesserextent, compared to the quality of life. Depression in patients with MS may be the result of struggling with the unpredictable nature of the disease, with daily uncertainty, lack of control over symptoms, and huge psychological tension [93], and is significantly related to how a person copes with the disease [94] and adapts to it. Of all the MS symptoms analysed, mood disorders have the greatest impact on the mental aspect of HRQoL. Other studies have also shown that the effects of depression on HRQoL are significant even after taking into account the effects of mediating variables, such as disability, fatigue, and cognitive disorders [16]. Even mild symptoms of depression have a significant impact on HRQoL, and therefore depression was identified as one of the strongest predictors of low HRQoL in individuals with MS, which affects all aspects of this disease [53,62,73,76].

Depression can influence HRQoL in patients by affecting other psychological variables, such as perception of social support, self-esteem, or sense of control over their health. It can also affect HRQoL directly, causing fatigue, memory problems, and affecting concentration. The third explanation points to the inverse relations: patients who rate their HRQoL as low are more likely to experience depression [56]. Depression occurs most often when all an individual’s resources and coping skills are exhausted. It affects one’s perception of the world and of one’s self, including one’s state of health, which may be judged more negatively than it actually is. Depression reduces motivation and adversely affects a patient’s physical progress during rehabilitation. In addition, factors that affect HRQoL negatively can also influence mood [37].

Some studies indicate a strong correlation between depression and physical disability [17,62]; nevertheless, some do not reveal such a connection [95,96]. In addition, researchers also found a link between fatigue and depression [15]. Mood disorders in people with MS may also be associated with cognitive disorders [62]. Among psychosocial factors, the increased risk of depression is associated with uncertainty about the future, helplessness, lack of hope, ineffective coping strategies (focused on emotions and avoidance), poor social relationships, inability to engage in recreational activities, and high levels of stress and fatigue. Some authors indicate that social situation has a greater effect on the symptoms of depression than physical disability [93].

Despite the high incidence of depression in people with MS, this symptom is diagnosed too rarely, and most patients do not have access to appropriate treatment, which is particularly important given that depression is a significant risk factor for suicidal ideation and suicide attempts in individuals with MS [17,20,21,24,28,60,62,80]. It is worth noting that among MS patients, there is a significantly higher proportion of suicides compared to the general population and individuals with other chronic diseases. The risk of suicide among MS patients is 5–10 times higher than in the general population. Suicidal thoughts are had by 28.6% of patients, and the suicide rate is 6.4%. The main risk factors for suicide are depression, alcohol abuse, and social isolation [28,80]. Depression is also often associated with anxiety, which is also a factor that reduces HRQoL, but it is much less frequently investigated than depression or cognitive disorders. The fear of disease progression is the most onerous for individuals with MS [34]. Additionally, other neuropsychiatric symptoms, such as agitation, irritability, apathy, and behavioural disorders, may affect HRQoL and adaptation to multiple sclerosis [97].

Adaptation to illness, assessment of HRQoL, and its mental and physical aspects are also related to other problems occurring in the course of MS, such as chronic pain, nausea, and imbalance. They are, like mood disorders, strongly associated with HRQoL and, to a lesser extent, with adaptation to multiple sclerosis. Chronic pain is increasingly recognized as a common problem among patients with multiple sclerosis. It is believed that about 50% of MS patients struggle with it [68]. Adaptation to chronic pain is primarily related to an individual’s ability to change behaviours and cognitive processes to relieve the experienced pain and its effects on other aspects of life. Fatigue, pain, and sleep quality are often underestimated symptoms of MS, which have a significant impact on the quality of life yet are often not included in scales which study MS, such as EDSS [15].

The last symptom associated with both adaptation to MS and HRQoL is upper-limb disability. This symptom is related to HRQoL much more closely, in particular to its physical aspect, as compared to adapting to MS. Upper-limb disability includes paralysis, spasticity, disorders related to cerebellar damage, ataxia, and tremors. Disability of the upper limbs contributes to the deterioration of the overall functioning of the patient; affects the ability to perform daily activities, such as eating, drinking, or writing; and thus reduces the quality of life of an individual and can affect their adaptation to MS.

The remaining MS symptoms (i.e., cognitive disorders, vision problems, speech impairment, swallowing problems, bladder dysfunctions, intestinal dysfunctions, and sexual problems) are not related to adaptation to illness. A particularly striking result is the lack of correlation between adaptation to illness and cognitive disorders. Analysing the role of individual factors affecting adaptation to chronic diseases, many authors often point to the role of cognitive processes [13,98]. However, patients may not be fully aware of their cognitive problems. They most often complain about slowing down, being distracted, and memory problems but may not be aware of disturbances of executive and visuospatial functions [99]. In addition, cognitive problems can also affect the perception of one’s own health and make patients less aware of their condition; this could be the reason why no correlation between cognitive and adaptive problems was found in this study.

However, it was found that there was a relation between cognitive disorders, HRQoL, and its physical and mental aspects. Cognitive disorders correlate very strongly with the mental aspect of HRQoL. Numerous authors point out that cognitive disorders are observed in a large number of patients [100,101,102]. These already appear in the early stages of the illness and tend to progress with time [103]. Cognitive deficits occur in MS even in the absence of physical disability [104]. Cognitive disorders in individuals with MS are persistent; difficult to treat; and associated with frustration, stress, and mood disorders [105]. Remission of cognitive symptoms is rare [106,107].

Patients with cognitive disorders are more likely to be unemployed, have limited social activity, and are also more dependent on their caregivers [16]. It has been shown that impairment of executive functions and memory dysfunctions are associated with deterioration in the HRQoL of individuals with MS [106,107,108,109], but there are exceptions that do not confirm this relation [110]. It was found that patients with autobiographical memory disorders reported higher HRQoL [111]. This may be due to the fact that patients with cognitive disorders have worse insight into their own health, which reduces the impact of disease progression on HRQoL [97,112,113].

Another symptom associated with HRQoL and its physical and mental aspects is intestinal dysfunctions. This relates to the physical aspect rather than the mental one. Although intestinal problems receive less focus than urinary disorders, this study has shown that they have a greater impact on patients’ mental health. Norton and Chelvanayagam [114] found that 34% of patients struggled with bowel disorders for more than 30 minutes a day. These authors point out that intestinal dysfunctions have a similar impact on HRQoL as physical disability related to mobility problems. Constipation and incontinence are two kinds of problems with the functioning of the intestines. Constipation and related physical problems—discomfort, abdominal distension, and pain—affect well-being, daily life, social functioning, and individual activity [115,116,117]. Studies show that constipation affects HRQoL assessment negatively and overlaps with disorders in other areas of functioning [118]. Faecal incontinence has an even greater impact on HRQoL than constipation; it occurs most often in individuals who do not move on their own [119]. Faecal incontinence is associated with significant levels of stress, as it is an embarrassing symptom, limiting activity and strongly affecting the functioning of patients, which may lead to social isolation [120,121]. It is associated with mental distress, shame, and a high level of uncertainty, as it does not appear regularly [119,122] and the associated lack of control leads to it dominating an individual’s activity [123]. The daily functioning of those affected is accompanied by the fear of getting dirty, fear of having no access to a toilet, and the necessity of having spare clothes to change [124]. It is not surprising that in patients with faecal incontinence, psychiatric problems—primarily mood disorders—are more frequent than in those without this disorder [125]. In addition, faecal incontinence is itself associated with the image of a person with mental problems or retardation [124], and due to the sense of shame and embarrassment, this problem is often hidden from family and friends. Therefore, many authors point out that faecal incontinence is a symptom that affects the HRQoL of individuals with MS strongly [121,123,125,126]. Another symptom often associated with intestinal dysfunctions is bladder problems, the most serious of which is urinary incontinence. In the current study, this problem correlated with HRQoL and its physical aspect as strongly as intestinal dysfunctions. However, it was not related to the mental aspect of HRQoL. The inability to control the bladder is a very stressful symptom which negatively affects mental and social functioning [119]. Urinary incontinence can isolate the patient from their environment and affect their daily activities. Some patients with MS use the toilet dozens of times a day, which makes them unable to leave home [118]. Bladder dysfunctions can be unpredictable and can cause sleep deprivation, exhaustion, fatigue, and even an increase in disability [127]. As urinary incontinence is a cause of isolation, depression, and shame and is related to many emotional problems, it has a negative impact on the HRQoL of patients, which has been demonstrated in many studies [127,128,129].

Urinary incontinence may also have a negative effect on sexual functions, which is another symptom of multiple sclerosis which affects HRQoL [126,130]. The current study found a moderate relation between sexual dysfunctions, HRQoL, and its physical aspect. As for the mental aspect, it was affected neither by sexual problems nor bladder dysfunctions. Although sexual dysfunctions are often underestimated compared to other MS symptoms, since multiple sclerosis usually occurs in young people, they may have a negative impact on patients’ functioning, quality of life, and close interpersonal relationships [131]. Many studies have found that the presence of sexual dysfunctions affects various aspects of HRQoL [117,130,132]. The decreased quality of life of individuals with MS who have sexual dysfunctions is independent of illness progression and degree of disability [133]. A relationship between sexual dysfunctions and bladder problems has been found in many studies [118,132], as has one between urinary incontinence and sexual life, which affects HRQoL negatively [76,126,132,134].

HRQoL is also affected by speech problems. In the current study, they were related both to overall HRQoL score and to its physical and mental aspects. Speech problems are related to a greater extent to the mental aspect of HRQoL. In the literature, the impact of speech disorders on the HRQoL of individuals with multiple sclerosis is rarely discussed. It can be assumed that the impact of this problem on HRQoL is related to the ability to communicate freely and its impact on an individual’s social situation.

The study also found a weak relation between impaired vision, HRQoL, and its physical aspect. This was confirmed in other studies, which found that visual impairment may affect HRQoL negatively in individuals with MS [135]. Impaired swallowing is also weakly related to HRQoL. However, in contrast to visual impairment, this problem affects HRQoL and its mental aspect yet isunrelated to its physical aspect. Dysphagia is a symptom that can lead to malnutrition [136], and its complications may be dehydration and septic pneumonia, which is the most common cause of death in MS [137]. This study found that impaired swallowing decreases the HRQoL of MS patients [138].

In recent years, HRQoL scales have been included in the study of the efficacy of disease-modifying drugs [37]. A number of studies showed that access to treatment is an important factor affecting HRQoL in MS patients. Therefore, the current study hypothesized the effect of disease-modifying treatment on adaptation to illness and HRQoL, but the hypothesis was not confirmed.

Other studies have demonstrated that patients treated over a longer period of time (more than fouryears) have a higher HRQoL score than those who were treated for shorter periods of time [139], and that disease-modifying drugs affect HRQoL [56,140]. The increase in HRQoL score was not related to age, duration of illness, disability, or number of previous relapses. Patients with the lowest HRQoL scores at the beginning of the study benefitted most from the administered treatment [140]. Several studies on the effects of interferon beta on HRQoL were published. In some, no relationship was found, while others showed a significant effect on HRQoL [15,141,142,143]. Studies indicate that administering treatment with interferon beta slows the disease process and improves HRQoL. This therapy is beneficial for patients’ HRQoL in spite of the occurrence of side effects similar to flu symptoms [54]. Also, the administration of natalizumab (Tysabri) improves HRQoL [75]. The impact of rehabilitation on HRQoL was also investigated, but it was found that it had a small impact on improving the assessment of HRQoL [144].

In summary, not all variables associated with the course of multiple sclerosis were associated with adaptation to the illness and health-related quality of life. Time since diagnosis, age at diagnosis, and treatment type did not have an influence on adaptation levels or quality of life. Both motor disability measured by EDSS and neurological disability measured by GNDS turned out to have a significant influence on adaptation to illness and quality of life. Adaptation to illness was associated with only five symptoms of multiple sclerosis, and the strongest relationships were observed with lower-limb disability and fatigue. Health-related quality of life was associated with all symptoms of multiple sclerosis; the strongest relationships were observed for both lower- and upper-limb disability, fatigue, other problems related to the course of multiple sclerosis, and mood problems. Mood problems, fatigue, and cognitive impairment were the strongest correlates of the psychological aspect of health-related quality of life.

## Figures and Tables

**Table 1 ijerph-15-02678-t001:** Health-related quality of life in individuals with multiple sclerosis (MS).

Variable	N	Minimum	Maximum	Average	Standard Deviation	Coefficient of Variation
Quality of life	137	29	132	75.37	26.04	34.55
Physical aspect	137	20	97	51.62	19.33	37.46
Mental aspect	137	9	43	23.74	9.48	39.91

**Table 2 ijerph-15-02678-t002:** Differences in the level of adaptation to illness and health-related quality of life in patients with various types of multiple sclerosis (ANOVA).

Descriptive Statistics				Levene’sTest		ANOVA	
Variables	Type	M	SD	F	*p*	F	*p*
Acceptance of illness	Relapsing–remitting	24.93	8.60	0.24	0.789	0.34	0.713
	Secondary progressive	23.18	8.05				
	Primary progressive	24.71	8.21				
	Total	24.46	8.30				
HRQoL physical aspect	Relapsing–remitting	43.63	18.78	1.35	0.265	8.46	0.001 ***
	Secondary progressive	61.14	15.26				
	Primary progressive	57.00	19.34				
	Total	51.96	19.62				
HRQoL mental aspect	Relapsing–remitting	21.60	9.84	2.53	0.085	1.22	0.299
	Secondary progressive	23.00	7.09				
	Primary progressive	24.94	9.07				
	Total	23.00	9.06				
HRQoL	Relapsing–remitting	65.23	26.23	2.16	0.122	5.96	0.004 **
	Secondary progressive	84.14	18.66				
	Primary progressive	81.94	26.95				
	Total	74.96	26.25				

HRQoL: health-related quality of life; ** *p* < 0.01, *** *p* < 0.001

**Table 3 ijerph-15-02678-t003:** Biomedical variables, adaptation to illness, and health-related quality of life.

Variable	Adaptation to Illness	HRQoL	HRQoL—Physical Aspect	HRQoL—Mental Aspect
Duration of illness	0.134	0.106	0.143	0.000
Age at the time of diagnosis	−0.006	−0.085	−0.089	−0.052
Motor impairment EDSS	−0.308 **	0.551 **	0.646 **	0.197 *
Neurological disability GNDS	−0.329 **	0.677 **	0.657 **	0.519 **

EDSS: Extended Disability Status Scale; GNDS: Guy’s Neurological Disability Scale; * *p* < 0.05; ** *p* < 0.01.

**Table 4 ijerph-15-02678-t004:** Assessment of self-mobility, use of rehabilitation equipment, and need for treatment which modifies the course of MS versus adaptation to illness and health-related quality of life.

Variable	Adaptation to Illness	HRQoL	HRQoL—Physical Aspect	HRQoL—Mental Aspect
Assessment of self-mobility (1-yes, 2-no)	−0.138	0.364 **	0.442 **	0.098
Use of rehabilitation equipment to move (1-yes, 2-no)	0.227 **	−0.458 **	−0.561 **	−0.112
Need for treatment which modifies the course of the illness (1-yes, 2-no)	0.040	−0.042	−0.035	−0.044

HRQoL: health-related quality of life; ** *p* < 0.01.

**Table 5 ijerph-15-02678-t005:** Symptoms of MS and adaptation to illness versus the health-related quality of life.

MS Symptoms	Adaptation to Illness	HRQoL	HRQoL—Physical Aspect	HRQoL—Mental Aspect
Cognitive disorders	−0.059	0.313 **	0.223 **	0.426 **
Mood disorders	−0.250 **	0.426 **	0.313 **	0.523 **
Impaired vision	−0.143	0.180 *	0.178 *	0.160
Impaired speech	−0.073	0.236 **	0.174 *	0.316 **
Impaired swallowing	0.047	0.179 *	0.157	0.183 *
Upper-limb disability	−0.191 *	0.459 **	0.524 **	0.207 *
Lower-limb disability	−0.324 **	0.508 **	0.624 **	0.141
Impaired bladder functions	−0.092	0.324 **	0.398 **	0.113
Impaired intestinal functions	−0.166	0.349 **	0.333 **	0.282 **
Sexual problems	−0.073	0.273 *	0.293 **	0.111
Fatigue	−0.314 **	0.499 **	0.446 **	0.461 **
Other problems	−0.242 **	0.442 **	0.410 **	0.350 **

MS: multiple sclerosis; HRQoL: health-related quality of life; * *p* < 0.05; ** *p* < 0.01.

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
