# Peer review of "Biomedical Variables and Adaptation to Disease and Health-Related Quality of Life in Polish Patients with MS"

_ijerph, 2018, doi:10.3390/ijerph15122678_

Round 1

Reviewer 1 Report

Comments on the Authors' decisions. I suggest removing the results (correlations) from the text, because they are included in the tables. In Table 2 no significance (stars) markings were inserted. I do not have other important remarks. The text refers to clinically important issues, the results described in detail. 

Author Response

Dear Sir or Madame, 

we would like to you use this opportunity to thank you for your suggestions and revisions. We did our best to incorporate your recommendations to this manuscript.

We have removed the results regarding correlations from the text, and now they are only presented in the tables as well we have added the stars.

Thank you again, all suggestions have been included.

Best wishes!

Reviewer 2 Report

The authors explore links between adaptation to disease and clinical disability and quality of life. The study and the data are well done but the text makes it hard to understand and the adaptation concept could be better explained. There are a number of other issues I noticed that would make the paper stronger if fixed:

The references appear duplicated each time

Line 30: "By destroying myelin in the brain and spinal cord, MS leads to permanent disability..." should be changed because the myelin can regenerate resulting in a relapsing remitting disease course in RRMS. When its ability to regenerate is exhausted the disability is permanent. Permanent disability is usually attributed to axon loss.

Line 49: explain the term sexually-adjusted (is it a statistical adjustment, or a behavior characteristic)

The first paragraph introduction needs to explain the concept of adaptation to disease. Is this a new idea of the authors, or an established paradigm?  The actual questionnaire appears to determine "acceptance of illness" which could be interpreted rather than adaptation to  disease. Please introduce the terminology for those unfamiliar with jargon.   

There are typos throughout e.g. Line 71  , line 299

Section 4 should address the patients treatment. Are they on a drug, have they recently had steroid treatment. Also, were they in remission at the time of filling the questionnaire for RRMS? Were they alone or allowed to be with a caregiver or family member during the interview? This could affect the accuracy of the self-reporting. Were other co-morbitities excluded

Line 99 what defines a medium vs.large city

Line 89: "The study consisted of the completion of a set of questionnaires..." however, it says later they also had clinical and psychological interviews, and their medical records were reviewed.Did the patients consent to the interview and medical record review?

Line 123: how were the cutoff determined, or was it arbitrary.  

multiple sclerosis is spelled out in several places, should use acronym MS throughout

It is hard to tell where the results section begins, is it part 6? A sub-title would help

The results text does not always match the table for example, line   204  "lower-limb disability (r = -0.34 " but in the table 5 it says -0.324 for this correlation. Need to double check the text results against the table results for accuracy.

The discussion is difficult to follow and it is a very dense at 5 pages. Suggest editing and trimming the discussion for flow and focusing on key points of adaptation to illness/disease concepts.

line 302 "contrary to what physicians claim" this is subjective may not apply to all physicians

line 357 " some do not reveal such a connection" but there are no papers referred to for this idea.

There is a sample bias for cognition since patients with severe cognitive disability were excluded. So the statement line 365 "A particularly striking result is the lack of correlation between adaptation to illness and cognitive disorders." should be qualified with mentioning the sample bias. 

Table 3 does not appear to have EDSS and GNDS although the text says it does. Perhaps the terms motor impairment and neurological disability are the same thing as EDSS and GNDS, but it is not clear for the reader.
